# Design, Synthesis, and Biological Evaluations of a Novel Resveratrol-Type Analogue Against VEGF

**DOI:** 10.3390/molecules30112345

**Published:** 2025-05-27

**Authors:** Shengying Lin, Maggie Suisui Guo, Roy Wai-Lun Tang, Yutong Ye, Jiahui Wu, Yuen Man Ho, Ran Duan, Ka Wing Leung, Tina Ting-Xia Dong, Karl Wah-Keung Tsim

**Affiliations:** Center for Chinese Medicine R & D, Division of Life Science, The Hong Kong University of Science and Technology, Clear Water Bay, Kowloon, Hong Kong, China; lishlin@ust.hk (S.L.); maggieguo@ust.hk (M.S.G.); roytwl@ust.hk (R.W.-L.T.); yyebl@connect.ust.hk (Y.Y.); jwuct@connect.ust.hk (J.W.); janetho@ust.hk (Y.M.H.); duanran@ust.hk (R.D.); lkwing@ust.hk (K.W.L.); botina@ust.hk (T.T.-X.D.)

**Keywords:** VEGF inhibitor, SAR study, resveratrol-type analogs, angiogenesis, age-related macular degeneration, cancer

## Abstract

Vascular endothelial growth factor (VEGF), also known as VEGF-A, has been reported to mediate various diseases, including cancer and wet age-related macular degeneration (wAMD). Despite the fact that VEGF inhibitors are commercially available and appear to be effective in clinical applications, adverse effects have been caused by these treatments. There is an unmet need for developing novel VEGF-targeted treatments against these diseases. Resveratrol, a phytochemical derived from fruits and vegetables, has shown promising potency in suppressing VEGF-mediated bioactivities through a series of in vitro and in vivo testing models. Herein, we report that RE-1, a synthetic resveratrol-type analog, displays robust inhibitory activities against VEGF and its downstream signaling pathways, surpassing its parental molecule, resveratrol. In addition, the drug capabilities of RE-1 were evaluated. As a newly synthesized chemical, RE-1 could be considered for subsequent pharmacological development targeting VEGF-related diseases.

## 1. Introduction

The vascular endothelial growth factor (VEGF) family is a subgroup of platelet-derived growth factors (PDGFs) and is characterized by a complex of three intramolecular disulfide bridges and the formation of homodimers through a cysteine knot motif [1]. The VEGF family consists of several members, including VEGF-A, VEGF-B, VEGF-C, VEGF-D, and so on, of which VEGF-A appears to be the most extensively investigated protein within the VEGF family. Interestingly, VEGF-A contains a group of isoforms, e.g., VEGF_110_, VEGF_121_, VEGF_145_, VEGF_162_, VEGF_165_, and VEGF_206_, and VEGF_165_ is recognized as the most abundant isoform of VEGF-A in nature [2,3].

The binding between VEGF proteins and VEGF receptors (VEGFRs) initiates various signaling pathways, activating the tyrosine kinase domain of the corresponding receptors and resulting in diverse biological activities, e.g., cell proliferation, migration, growth, and apoptosis [4,5]. Metastatic tumor growth requires fundamental nourishment and oxygen, which are supplied by the development of autonomous blood vessels through angiogenesis. Intriguingly, VEGFs play a crucial role in mediating angiogenesis and have been implicated in the progression of various cancers, such as breast and colorectal cancer [6]. In addition, VEGF-regulated angiogenesis has also been linked to wet age-related macular degeneration (wAMD), an eye disease usually triggered by the production of choroidal neovascularization, with the retinal pigment epithelium contributing to AMD pathogenesis [7,8]. Thus, anti-VEGF therapeutics could be a beneficial strategy in drug development against cancer and wAMD.

Targeted therapy has been employed as an efficient tool for designing novel ligands against various diseases [9]. In the 1890s, Paul Ehrlich proposed the principle of targeted therapy as a “magic bullet” that would be utilized to specifically interact with targeted proteins without triggering unexpected toxicity [9,10]. Since then, the concept has been applied to tackle infectious diseases and cancer through agents such as trastuzumab and imatinib, which target HER2-mediated breast cancer and chronic myelogenous leukemia, respectively [9,10,11]. This inspired us to design a series of novel inhibitors targeting the VEGF protein to block downstream signaling pathways and treat VEGF-mediated diseases.

Resveratrol, a phytochemical derived from fruits and vegetables, has shown anti-angiogenic and anti-cancer effects by inhibiting VEGF and disrupting the VEGF/VEGFR1 signaling pathway [12,13,14]. Such a phytochemical could serve as a hit molecule for subsequent structure–activity relationship (SAR) studies aimed at improving its efficacy and toxicity profile. We found the *para*-phenol scaffold of resveratrol to be synthetically attractive, which enabled us to introduce various building blocks through this vector (Figure 1A), potentially yielding a group of novel molecules with significant binding affinities to VEGF and potent anti-VEGF activity. A group of resveratrol-type derivatives were designed, synthesized, and subjected to biological evaluations through diverse bioassays. Here, we reveal that the newly synthesized chemical, RE-1, exerts promising inhibitory activity against VEGF protein and appears to be a promising anti-VEGF agent. Intriguingly, RE-1, in comparison, demonstrates higher potency than its parental molecule, resveratrol, and exhibits lower toxicity in cell studies, suggesting its potential for further pharmaceutical development.

## 2. Results

### 2.1. RE-1 Expresses Good Binding Affinity to VEGF Protein

The chemical structure of RE-1 consists of a resveratrol fragment and a four-membered ring building block, connected via a propane bridge to, designed to occupy the active site of VEGF and enhance the ligand binding affinity. The propane bridge was introduced using a dichloropropane reagent through S_N_2 reactions in two steps [15]. As anticipated, several side products were observed in the first S_N_2 reaction, and intermediate **2** was collected as a crude mixture before being subjected to another S_N_2 reaction (Figure 1A,B). After purification using chromatography, RE-1 of good purity was (>95%), albeit only a 26% yield (over two steps) was recorded.

A computational docking study was conducted to predict the binding activity between the RE-1 and VEGF protein. Axitinib, a selective tyrosine kinase inhibitor [16], was utilized as a positive control, while resveratrol was employed as an internal control, both of which released −17.4 kJ/mol and −19.1 kJ/mol in binding to VEGF protein, respectively. As shown in Figure 1C,D, RE-1 anchored at the active site of VEGF and prevented the protein from binding to its receptor. In comparison, RE-1 required less energy (−25.3 kJ/mol vs. −19.1 kJ/mol), and thus should display a higher affinity to VEGF than resveratrol. This suggests that RE-1 could potentially be utilized as a more promising anti-VEGF inhibitor than resveratrol. A 2D protein−ligand diagram revealed that RE-1 bound to the same region as resveratrol and established multiple interactions with surrounding residues, such as pi−pi interactions with Phe36, covalent bonding with Pro40, and H-bonding with Asp63 (Appendix A).

In parallel, an ultrafiltration assay was conducted in the presence and absence of VEGF protein (Figure 2A). As a result, RE-1 showed a ~32% binding degree to the protein, slightly higher than that of resveratrol (~25%). Subsequently, SPR investigation was performed using Biacore T200 (Cytiva, Washington, DC, USA) to detect the binding affinities of RE-1 to the targeted protein. VEGF_165_ protein was placed onto the sensor chip while various concentrations of analytes, i.e., RE-1 and resveratrol, flowed through the chip. When the binding activities between analytes and the protein were detected, response units (RUs) could be calculated and obtained accordingly. As shown in Figure 2B and Appendix A, both ligands displayed decent affinities to the protein in dose-dependent manners from 1 to 30 µM. It is worth noting that the dissociation constant (K_d_) of RE-1 and resveratrol in this assay were estimated to be about 8 µM and 12 µM, respectively. This indicates that RE-1 appeared to interact more favorably with VEGF protein than resveratrol, which was very much in line with our observations in computational studies.

### 2.2. RE-1 Inhibits VEGF-Mediated Bioactivities

In light of the investigation above, we further evaluated the anti-VEGF efficacy of RE-1 in cell studies. VEGFR2 was previously identified as the main receptor of VEGF protein. We conducted a Western blot assay and revealed that both RE-1 and resveratrol were able to disrupt the phosphorylation and binding activity of VEGF/VEGFR2 complex (Figure 3). ERK1/2, a downstream signal factor, was subsequently stimulated and phosphorylated after the activation of VEGFR2 [17]. Both RE-1 and resveratrol were found to block ERK1/2 phosphorylation in the presence of VEGF (Figure 4). This suggests that RE-1 and resveratrol were likely to attenuate VEGF-mediated bioactivities through blockage of the VEGF/VEGFR2 complex.

VEGF has been reported as a crucial mediator in human umbilical vein endothelial cells (HUVECs). Indeed, VEGF was found to enhance the cell proliferation of cultured HUVECs (Figure 5) [18]. As expected, resveratrol expressed decent inhibition to VEGF-regulated cell proliferation, which was in agreement with our previous investigation. Intriguingly, RE-1 also displayed a significant inhibitory activity to the cell viability of HUVECs in a dose-dependent manner when co-treated with VEGF protein. In addition, we employed a wound closure assay in HUVECs and, as shown in Figure 6, an efficient wound healing assay was observed from the control group (VEGF only). Resveratrol and RE-1 both showed significant inhibitions to wound closure by inhibiting VEGF in a dose-dependent manner, of which RE-1 achieved a higher efficacy than resveratrol from 0.3 µM to 3 µM. This implies that, at relatively low concentrations, RE-1 could display a higher efficiency in the reduction of VEGF-mediated wound recovery than resveratrol.

VEGF acts as an inducer in reactive oxygen species (ROS) formation; such an interaction plays an essential role in the activation of VEGFR2-mediated pathways and downstream cell migration. Hence, it is believed that the inhibition of VEGF protein could lead to decreased ROS formation followed by the suppression of cell proliferation [19]. Initially, VEGF was shown to induce ROS formation without any treatments, and avastin (bevacizumab) was able to reduce such an induction from VEGF protein. Interestingly, both resveratrol and RE-1 expressed inhibitory effectiveness to VEGF-mediated ROS formation, of which RE-1 showed better inhibitory performances than resveratrol from 0.3 to 3 µM (Figure 7), which was in line with observations from the HUVECs studies above.

### 2.3. Pharmacological Properties of RE-1

To determine the pharmacological properties of RE-1, a series of detecting assays were established to obtain a pharmacological profile of this new chemical. An MTT assay was performed to investigate the potential toxicity caused by RE-1 in cultured HUVECs and HaCaT cells (Figure 8). As a result, no significant apoptosis was observed from RE-1 at concentrations up to 100 µM, in contrast with resveratrol, which triggered mild apoptosis at relatively high concentrations. Furthermore, we employed SEESAR Optibrium software (version 14.0) to generate a group of predictive ADMET statistics of RE-1 and resveratrol. As summarized in Appendix A, RE-1, strictly following Lipinski’s rule of five [20], i.e., MW < 500 Da, Log *P* < 5, H-bond donor and receptor < 5, was unlikely to cause toxicity to cytochromes P450 enzymes (CYPs), and showed no negative impacts on the blood−brain barrier (BBB). Further, it was predicted that RE-1 could not trigger active transportation through P-glycoprotein (P-gp), and it could be absorbed through human intestinal (positive HIA value) [21,22]. Meanwhile, we conducted a Caco-2 permeability assay to detect the cell absorption of RE-1 and propranolol, a well-absorbed drug, which displayed a promising *P*_app_ value of 46.8 × 10^−6^ cm/s [23]. As a result, the *P*_app_ value of RE-1 ranged from 15.8 × 10^−6^ cm/s to 20.9 × 10^−6^ cm/s at concentrations from 0.3 µM to 3 µM (Table 1), which were predicted to exhibit good absorption and oral bioavailability according to Ahmed et al. [23]. Moreover, a stability test through HPLC was performed and it revealed that RE-1 remained relatively stable in acidic, neutral, and basic solutions, as RSD values from both inter-day and intra-days were lower than 10% (Table 2).

## 3. Discussion

Protein kinases play an important role in cellular functions that drive cell growth and proliferation [24], which have been of interest for decades in order to develop selective small molecule inhibitors, particularly as treatments of diseases. Many cancer patients suffer from common toxicities of traditional chemotherapy, due to high dosing and the lack of selectivity between healthy and tumorous cells. Compared to cytotoxic chemotherapy, which mainly inhibits cell replication, targeted therapy interferes with specific molecular pathways involved in tumor growth. This mechanism helps block cancer cell propagation while potentially reducing the adverse effects commonly associated with classic cytotoxic therapy [25,26]. To date, seven inhibitors targeting the VEGF/VEGFR pathway have been approved by the FDA [27,28,29]. Two have been approved as anti-cancer treatments, i.e., bevacizumab and ramucirumab, while six have been utilized to tackle ocular vascular disease, including ramucirumab, pegaptanib, brolucizumab, and others. Regarding small molecule inhibitors, 72 kinase inhibitors have been approved by the FDA, of which 13 drugs have been clinically applied to target VEGF/VEGFR-related diseases. Nevertheless, severe adverse effects have been constantly reported from these marketed kinase inhibitors, such as arrhythmias, heart failure, and cardiac ischemia, due to off-target toxicity. This inspired us to design and develop more potent VEGF-targeted drug treatments with a high efficacy and potentially low toxicity [27,28,29].

During our previous investigations, we established a series of detecting platforms to screen a library of anti-VEGF agents, of which resveratrol appeared as a promising inhibitor against VEGF-induced biological activities. Subsequently, resveratrol was found to significantly attenuate VEGF-mediated angiogenesis, cancer growth, and ROP (retinopathy of prematurity)-related symptoms [12,13,14]. Intriguingly, structural modifications around resveratrol have been well studied [30,31]. For example, Szczepanska et al. [30] synthesized a group of resveratrol analogs that showed stronger anti-cancer effects against lung cancer (A549) and colorectal cancer cells (HT29) compared to resveratrol. In addition, a group of novel resveratrol derivatives was synthesized and demonstrated more potent inhibitory impacts on MDA-MB-231 breast cancer cells [31]. It is worth noting that no significant toxicity was observed from these resveratrol analogs mentioned above. This suggests that resveratrol could serve as a chemical hit molecule for subsequent drug design efforts aimed at improving its efficacy and generating more potent lead candidates.

The “hit-to-lead” phase, which plays a crucial role in drug development, usually commences to optimize hit molecules towards “drug-like” lead candidates through SAR studies [32]. The fragment-based approach has been an efficient strategy in drug design and the “hit-to-lead” stage in order to develop “small fragments” into bigger molecules that form stronger interactions with the targeted proteins [33]. In this study, resveratrol was employed as a hit molecule and enabled us to introduce various “building blocks” into the fragment to yield highly efficacious candidates. Indeed, after several rounds of SAR studies, we designed and synthesized a group of resveratrol-type derivatives. Interestingly, compared with resveratrol, RE-1 displayed s higher binding affinity to the VEGF protein in computational docking and SPR detection, and expressed more potent effectiveness in disrupting the VEGF/VEGFR-1 signaling pathway and suppressing downstream bioactivities. This strongly suggests that RE-1 has been identified as a more effective anti-VEGF agent than resveratrol; therefore, it could be considered to be a promising drug candidate against VEGF-related diseases, although intensive in vivo studies, as well as pharmacokinetic and pharmacodynamic investigations, are still required in order to provide a pharmacological prolife for this novel resveratrol-type analog.

wAMD is an eye disease that affects elderly individuals and significantly impacts patients’ quality of life. Current treatments to tackle this disease have been limited to bevacizumab (avastin) or ranibizumab (lucentis) through intravitreal injection. This invasive administration could potentially lead to severe adverse effects, such eye bleeding and inflammation, and is a heavy burden to hospitals and patients due to frequent visits [34,35]. We previously revealed resveratrol as a potent anti-VEGF inhibitor that was able to attenuate the VEGF-mediated retinopathy of prematurity in eye-drop formulations [13]. As a resveratrol-type analog, RE-1 expresses a more significant inhibitory performance to VEGF-regulated biological activities than resveratrol; therefore, it could serve as a promising potential therapeutic against wAMD disease through topical instillation.

## 4. Materials and Methods

### 4.1. Chemicals and Reagents

All chemical reagents and solvents were purchased from commercial companies. NMR spectra (^1^H NMR and ^13^C NMR) was conducted using s Bruker (Boston, MA, USA) Advance 400 (^1^H: 400 MHz; ^13^C: 101 MHz). The abbreviations for spin multiplicity are as follows: s = singlet; d = doublet; t = triplet; m = multiplet. HRMS were provided through Agilent 6550 iFunnel QTOF (Santa Clara, CA, USA). Reagents used in this report, including Dulbecco’s modified Eagle medium (DMEM), fetal bovine serum (FBS), and other related reagents, were obtained from Thermo Fisher Scientific (Waltham, MA, USA). Various antibodies species were obtained from the Cell Signaling Technology company (CST; Beverly, MA, USA). MTT [3-(4,5 dimethy-2-thiazolyl)-2,5-diphenyl-2H-tetrazolium bromide] and DCFH-DA probe (20,70-dichlorofluorescein diacetate) were obtained from Sigma-Aldrich (St. Louis, MO, USA).

### 4.2. Synthesis of Chemicals

Synthesis of compound **2**: To a solution of amine 1 (1 eq.) in anhydrous THF (5 mL), NaH (60% in mineral oil, 1 eq.) was added slowly at 0 °C and the reaction mixture was allowed to stir at room temperature for 1 h before the addition of 1,3-dichloropropane (1 eq.) slowly. After 8 h at room temperature, the mixture was quenched with H_2_O (10 mL/mmol) and extracted with EtOAc three times. The combined organic layers were washed with brine, dried (MgSO_4_), filtered, and concentrated under a vacuum to produce a mixture that was processed by chromatography to produce compound **2** with impurities. HRMS detection for [M + H]^+^: found 391.2547.

Synthesis of RE-1: To a solution of pterostilbene (1 eq.) in anhydrous THF (5 mL), NaH (60% in mineral oil, 1 eq.) was added slowly at room temperature and the reaction mixture was allowed to stir at room temperature for 1 h before the addition of compound **2** (1 eq.). After 16 h at room temperature, the mixture was quenched with H_2_O (10 mL/mmol) and extracted with EtOAc three times. The combined organic layers were washed with brine, dried (MgSO_4_), filtered, and concentrated under a vacuum to produce a mixture that was purified by chromatography to produce RE-1 in 26% (combined step I and II). ^1^H NMR (400 MHz, DMSO-*d*_6_) δ 1.77–1.81 (m, 2H), 1.98–2.00 (m, 2H), 2.07–2.09 (m, 2H), 2.58–2.63 (m, 2H), 2.87–2.91 (m, 2H), 3.72–3.78 (m, 8H), 4.09–4.13 (m, 1H), 5.65 (s, 1H), 6.03 (d, J = 5.6 Hz, 2H), 6.67–6.75 (m, 4H), 6.89 (d, J = 6.4 Hz, 2H), 7.02 (d, J = 10.2 Hz, 1H), 7.20–7.22 (m, 2H), 7.23 (d, J = 10.2 Hz, 1H), 7.38 (d, J = 6.4 Hz, 2H), 7.72 (d, J = 2.2 Hz, 1H). ^13^C NMR (101 MHz, DMSO-*d*_6_) δ 18.9, 19.7, 24.0, 29.4, 50.7, 55.8 (2C), 58.9, 65.1, 100.2, 106.3 (2C), 109.6, 110.9 (2C), 112.8, 113.7, 115.0 (2C), 121.1, 122.8 (2C), 124.7, 126.5, 127.4, 128.9, 130.1 (2C), 131.5, 132.7, 134.2, 138.9, 158.4, 160.8 (2C). HRMS detection for [M + H]^+^: found 611.1875.

### 4.3. Cell Cultures

Cell lines employed in this report, including Human Umbilical Vein Endothelial Cell (HUVECs), human keratinocyte cell lines HaCaT, and human colorectal adenocarcinoma Caco-2 cell lines, were obtained from American Type Culture Collection (ATCC, Manassas, VA, USA). In the process of cell culturing, cell lines were supplemented with DMEM culture medium and subsequently supplied with 10% FBS and 1% penicillin/streptomycin (100 U/mL and 100 μg/mL). The cells were then maintained in a humidified CO_2_ incubator environment with conditions of 5% CO_2_ and 37 °C. After that, the cell lines were accordingly subjected to a subculture when the cell confluency exceeded 80%.

### 4.4. Binding Affinity Assay

Binding affinities between analytes and VEGF protein were determined using a Biacore T200 (Cytiva, Washington, DC, USA) with a GE Series dextran-coated (CM5) sensor chip. Specifically, the HBS-T mixture (150 mM NaCl, 10 mM Hepes, 0.05% polysorbate 20, 3.4 mM EDTA, 5% DMSO, pH 7.4) was selected as the operating buffer at 25 °C. The immobilizing VEGF protein was covalently placed onto the chip at the sensor surface through amide coupling reagent—EDC/NHS {1-ethyl-3-[3-(dimethylamino) propyl]carbodiimide hydrochloride)/N-hydroxysuccinimide}. Following surface activation, the VEGF protein was dissolved using a coupling buffer (0.1 M acetate buffer, pH 4.5) and placed at the surface of the sensor chip, until the RU (resonance unit) signal of the VEGF protein was detected and reached 6500 RU. To remove unreacted VEGF protein, the chip was washed and subsequently reacted with 10 mM glycine-HCl at pH 1.5. Various concentrations of resveratrol and RE-1 were diluted with a running buffer (tested concentrations were 1, 3, 10, and 30 µM) and flowed through the chip surface, respectively. The binding activities between the VEGF protein and tested ligands (either resveratrol or RE-1) were determined in real-time manners. A testing group without VEGF immobilizing protein served as a blank control. Data were further analyzed utilizing GE Biacore T200 control software.

### 4.5. Cell Viability

Cell viability was assessed with an MTT assay to evaluate the potential effect of the tested drugs on cells. Th cells were seeded onto 96-well plates for 24 h, prior to drug treatments. After incubations for 24 h, an MTT solution in a final concentration of 0.5 mg/mL was added to each well. Following another 3 h of incubation, DMSO solvent was used to dissolve the purple formazan crystals generated in each well. The absorbance of the samples at 570 nm was measured using a microplate reader (Thermo Fisher Scientific). The cell viability was calculated as follows: cell viability (%) = (experimental value − background value)/(blank value − background value) × 100%.

### 4.6. Wound Closure Assay

A wound closure assay was performed on cultured HUVECs to study cell migration in vitro. Procedures were described previously by Guo et al. [36]. Briefly, HUVECs were seeded onto a 12-well plate at a density of 3 × 10^5^ cells/well, and treatments were added when the cells grew to full confluence. The cells were washed with PBS and treated with drugs. The photos of each cell scrape at 0 (At_0_) and 20 h (At_20_) were captured with a microscope and imaging software (Zen, https://www.zeiss.com/microscopy/en/products/software/zeiss-zen.html, accessed on 19 January 2025) at a 10× magnification. The software TScratch 1.0 (CSE Lab, Zurich, Switzerland) was used to determine the open area of each scrape photo at various timepoints. Quantification of the recovery rate of wound was calculated according to the following equation: Wound closure (%) = (At_0_ − At_20_)/At_20_ × 100%.

### 4.7. Measurement of Intracellular ROS

Intracellular ROS accumulation was measured with a fluorescent ROS indicator DCFH-DA. Procedures were conducted as described previously [36]. In brief, HaCaT cells were seeded onto coverslips at a density of 1 × 10^5^ cells/mL. Following drug treatments, a DCFH-DA probe diluted in serum-free medium at 20 μM was introduced to the cells. The cells were incubated with DCFH-DA for 20 min in a 37 °C incubator. PBS wash was followed to rinse off the residual DCFH-DA. Following this, the cells were fixed with 4% paraformaldehyde (PFA) in PBS for 15 min at room temperature. The PFA was washed off with a PBS wash twice. Coverslips with fixed cells were mounted with Prolong Gold Antifade Reagent with DAPI (Cell Signalling Technology, Boston, MA, USA). Pictures of cells were captured with confocal microscopy Leica SP8 system at 20× magnification (Leica, Wetzlar, Germany). The intensities of fluorescent signals by DCFH-DA were determined through the software Leica Application Suite (LAS) X (Leica Microsystems, Wetzlar, Germany).

### 4.8. SDS/PAGE and Western Blotting

HUVECs were seeded onto 12-well plates at a density of 1.5 × 10^4^ cells/mL and subjected to serum starvation with a serum-free medium overnight. After treatment with various samples, the cells were directly lysed with 200 μL 2× sodium dodecyl sulfate polyacrylamide gel electrophoresis (SDS-PAGE) lysis buffer (100 mM Tris-Cl, pH 6.8, 8% SDS, 0.2% bromophenol blue, 20% glycerol, 400 mM β-mercaptoethanol). The lysates were obtained and denatured at 85 °C for 15 min. Afterwards, the samples were subjected to SDS-PAGE in a 10% gel. Then, 5% skim milk in Tris-buffered saline of pH 7.4 with 0.1% Tween-20 (as TBST) was used for blocking. Antibodies in use included: anti-rabbit phospho-p44/42 MAPK, anti-rabbit p44/42 MAPK (CST), and anti-rabbit horseradish peroxidase (HRP) secondary antibody (Sigma-Aldrich). ECL Western blotting substrate (Thermo Fisher Scientific) was applied to help visualize the blots with a Chemidoc Touch Imaging System (Bio-Rad Laboratories, Hercules, CA, USA). The intensities of each band were quantified and analyzed with software Image Lab.

### 4.9. Cell Proliferation

First, 5 × 10^4^ cells/mL HaCaT cells were seeded on a 96-well plate. After 24 h incubation, 100 μL VEGF at a concentration of 10 ng/mL with various concentrations of analytes were added to the mixture, followed by the addition of 10 μL of MTT solution (5 mg/mL). Th blank group was set up without any drug treatments, while avastin was utilized as a positive control. The mixture was incubated at 37 °C for 4 h before 150 μL DMSO was employed to dissolve formazan salt formed during the reaction. The sample was collected and read under a microplate reader (Thermo Fisher Scientific) at a wavelength of 570 nm. Proliferation values were calculated as follows: cell proliferation (%) = (experimental value − background value)/(blank value − background value) × 100%.

### 4.10. Computational Docking Study

Docking study was conducted as previously described [37]. Briefly, the chemical structures of RES and RE-1 were generated from Chemdraw (version 20.0, https://revvitysignals.com/products/research/chemdraw, accessed on 11 January 2025), and the VEGF protein structure was downloaded from the Protein Data Bank (PDB code: 1FLT, https://www.rcsb.org/, accessed on 11 January 2025). Residues 1–165 of VEGF were selected as the targeted domain in the binding stimulation. Virtual screening was performed using SEESAR software (Version 13.0; https://www.biosolveit.de/, accessed on 11 April 2024). The prediction of ADMET properties, including TPSA, Log *P*, intestinal absorption, and so on, was conducted under Optibrium mode.

### 4.11. Stability Test in HPLC

The stability test was conducted by determining the intra- and inter-day variability through HPLC analysis, which was performed on an Agilent 1200 Liquid Chromatography (Santa Clara, CA, USA) equipped with an ODS C18 column (4.6 mm × 250 mm). The analytes, resveratrol and RE-1, were dissolved in MeOH at 0.1 mg/mL. Resveratrol and RE-1 were both detected at a 210 nm wavelength. The intra- and inter-day stability were investigated by analyzing six replicated samples of the standard solution containing the two analytes during a single day and six replicates of the samples examined on five successive days, respectively. The relative standard deviation (RSD) was taken as a measurement of stability [37].

### 4.12. Trans-Epithelial Permeability of Chemical Transport

Caco-2 cells were cultured until fully differentiated, after being cultured for 21 days. The transepithelial permeability assay was conducted as reported [38]. The integrity of the cell monolayer was determined by the transepithelial electrical resistance (TEER) using an EVOM Epithelial Volt/Ohm Mete (WPI, Sarasota, FL, USA) and the permeability of lucifer yellow (a paracellular leakage marker) across the cell monolayer. Prior to drug treatments, the inserts were washed twice and equilibrated for 30 min with pre-warmed Hank’s balanced salt solution (HBSS, pH 6.0 at the apical side, pH 7.4 at the basolateral side). The transepithelial permeabilities of the analytes (RE-1 and RES) were measured using mass spectrometry. The *P*_app_ value was calculated as follows: $P_{\mathrm{app}}=\left(\frac{dQ}{dt}\right)\times \frac{1}{C}\times \frac{1}{A}$, where (*dQ/dt*) is the slope of cumulative concentrations of analytes (RES or RE-1) after being received over timepoints, and C is the initial concentration of analytes, while A is the surface area of the membrane, which is 1.12 cm^2^ here.

## 5. Conclusions

VEGF has been associated with various diseases such as cancer and wAMD. Despite the fact that a number of VEGF inhibitors have been approved by the FDA and are currently in use clinically, adverse effects have been constantly reported from these marketed drugs due to the lack of selectivity and off-target toxicity, which indicates that there is a considerable need to develop novel anti-VEGF agent with a higher efficacy. Resveratrol was previously revealed as a potent hit against VEGF, which enabled us to discover resveratrol-type analogs with improved potency. The newly synthesized RE-1 was able to exert better effectiveness than resveratrol through a series of in silico and in vitro studies. Such a derivative is worth further pharmacological investigations with the aim of delivering a significantly more effective anti-VEGF agent.

## Figures and Tables

**Figure 1 molecules-30-02345-f001:**
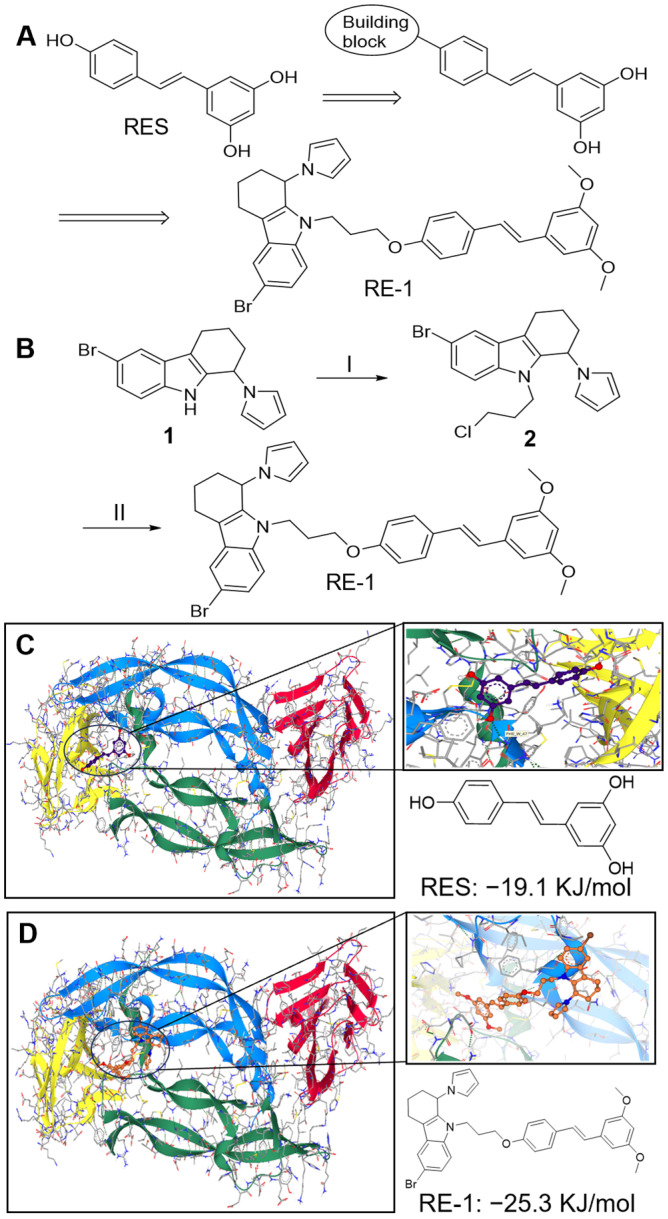
(**A**) Design of resveratrol-type derivatives. (**B**) Synthetic route towards RE-1. (I) 1,3-dichloropropane, NaH, THF, 0 °C-rt, 8 h; (II) pterostilbene, NaH, THF, rt, 16 h, 26%. Docking studies of RES (**C**) and RE-1 (**D**) against VEGF protein. VEGF_1–165_ (PDB code: 1FLT) domain was selected as a binding site and docking studies were performed using SEESAR software (13.0 version). RES: resveratrol.

**Figure 2 molecules-30-02345-f002:**
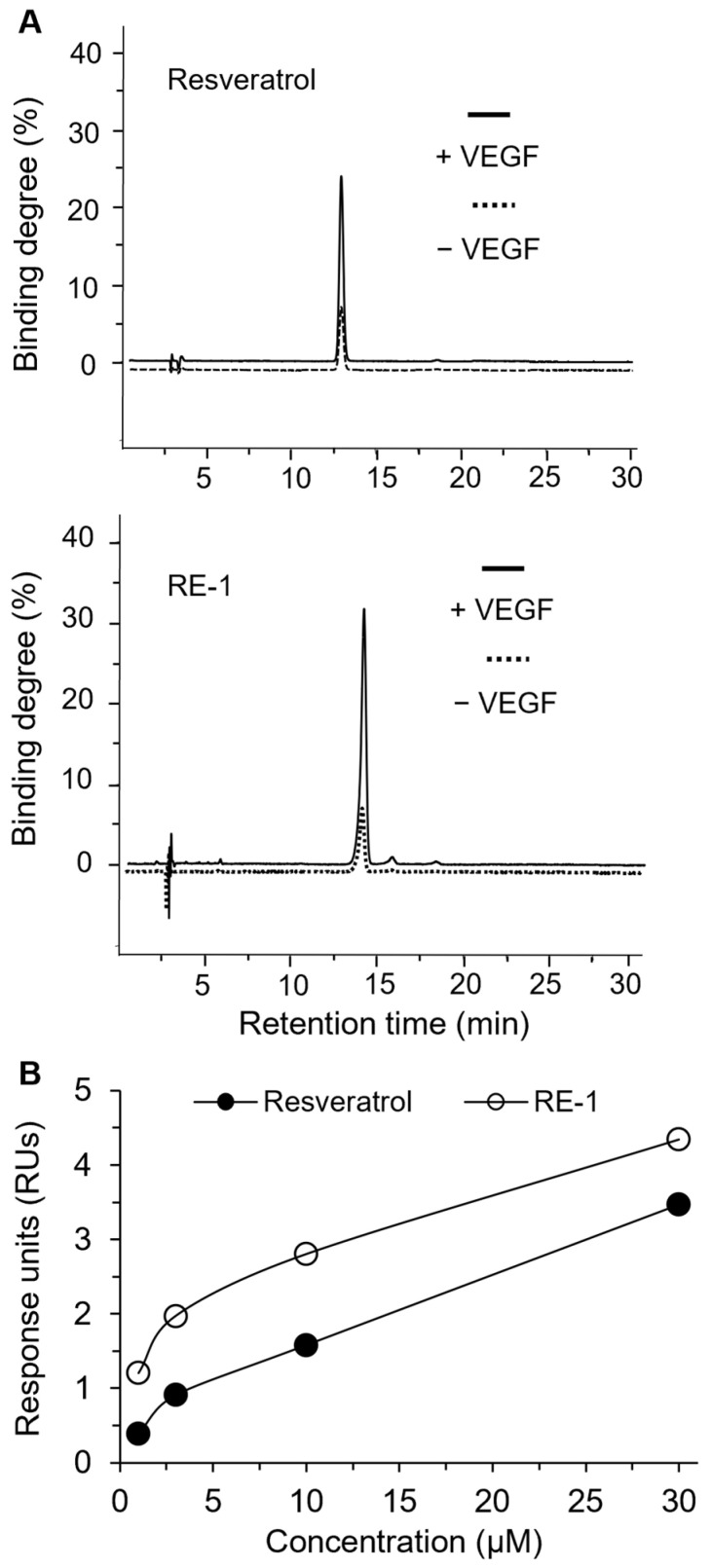
(**A**) Binding of RES and RE-1 with VEGF in assay using ultrafiltration. Quantification of phytochemicals was performed using HPLC. (**B**) SPR binding was performed using Biacore T200. Immobilized VEGF protein was placed onto a sensor chip, while analytes were flowing through the chip. RUs were recorded and calculated using GE Biacore T200 control software (version 3.2.2).

**Figure 3 molecules-30-02345-f003:**
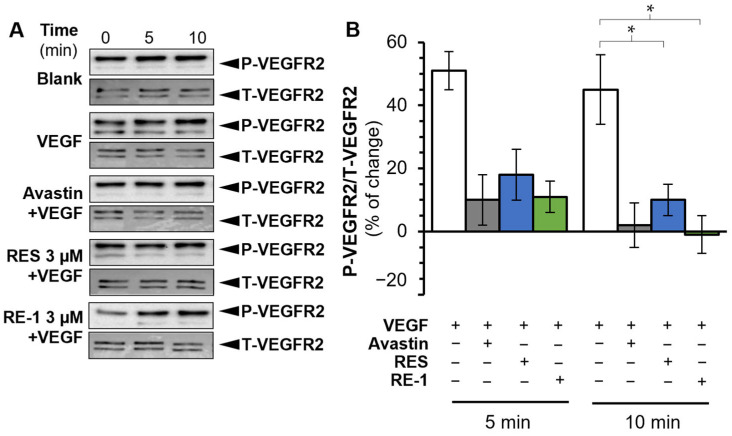
RE-1 and resveratrol disrupt the phosphorylation of VEGF/VEGFR2 in a Western blot assay. (**A**) VEGF was added at a concentration of 10 ng/mL, and avastin at 200 µg/mL was employed as the positive control. (**B**) The intensities of each band were quantified and analyzed with software Image Lab (6.1 version). Values were calculated as the percentage of changes compared with blank control (only medium), in mean ± SD, *n* = 4. * *p* < 0.05; compared with the control group.

**Figure 4 molecules-30-02345-f004:**
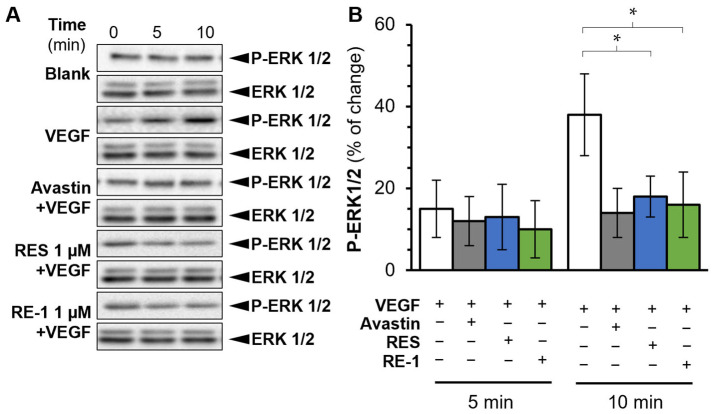
RE-1 and resveratrol inhibit phosphorylation of downstream ERK1/2 in a Western blot assay. (**A**) Cell treatment was the same as in Figure 3. RE-1 and resveratrol both displayed inhibitions to phosphorylation of ERK1/2. Antibodies in use included: anti-rabbit phospho-p44/42 MAPK, anti-rabbit p44/42 MAPK (CST), and anti-rabbit horseradish peroxidase (HRP) secondary antibody. (**B**) The intensities of each band were quantified and analyzed with the software Image Lab. Values were calculated as percentage of changes compared with a blank control (only medium), in mean ± SD, *n* = 4. * *p* < 0.05; compared with the control group.

**Figure 5 molecules-30-02345-f005:**
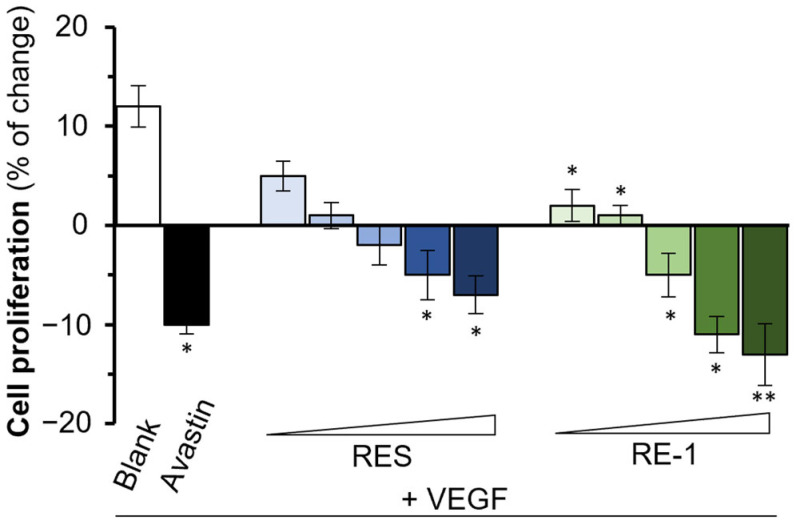
RE-1 and resveratrol attenuate VEGF-induced cell proliferation in HUVECs. HUVECs were incubated in the presence of VEGF (10 ng/mL) with or without drug treatments for 48 h. Both RE-1 and resveratrol were tested at concentrations of 0.1 µM, 0.3 µM, 1 µM, 3 µM, and 10 µM. The absorbance of the samples at 570 nm was measured using a microplate reader (Thermo Fisher Scientific, Waltham, MA, USA). * *p* < 0.05; ** *p* < 0.01; compared with the control group.

**Figure 6 molecules-30-02345-f006:**
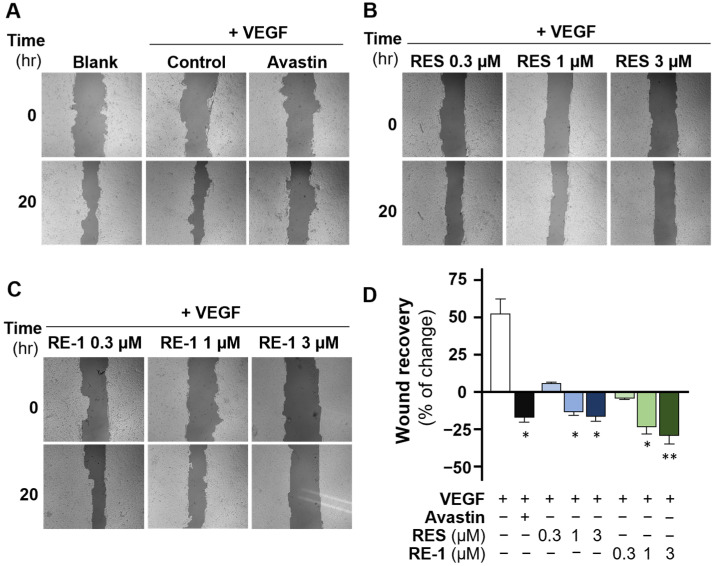
RE-1 reduces VEGF-regulated wound recovery in HUVECs. (**A**) Avastin (400 µg/mL) served as a positive control, and the treatment of VEGF (10 ng/mL) promoted wound healing in cultured HUVECs, while resveratrol (**B**) and RE-1 (**C**) inhibited the wound recovery from 0.3 µM to 3 µM. (**D**) Quantification of wound recovery. Values were calculated as percentage of changes compared with the blank control (only medium), in mean ± SD, *n* = 4. * *p* < 0.05; ** *p* < 0.01; compared with the control group.

**Figure 7 molecules-30-02345-f007:**
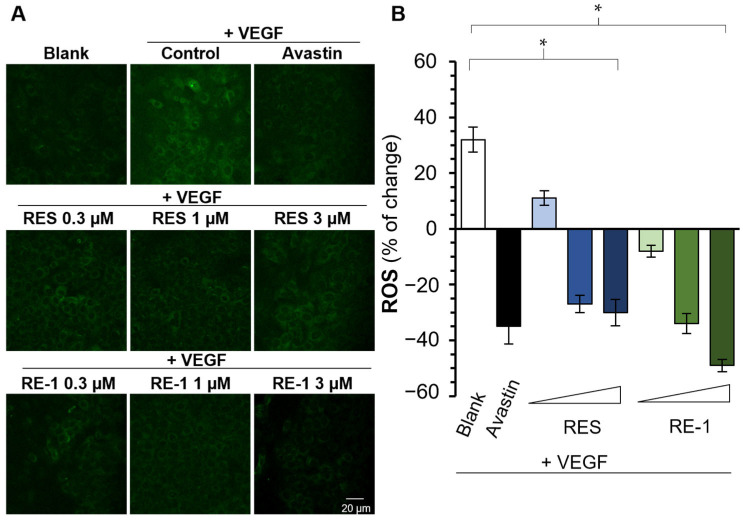
RE-1 attenuates VEGF-mediated ROS formation in HaCaT cells. (**A**) HaCaT cells were seeded onto coverslips and mounted with Prolong Gold Antifade Reagent with DAPI. Pictures of cells were captured with confocal microscopy Leica SP8 (Leica, Germany). (**B**) The intensities of fluorescent signals by DCFH-DA were quantified using the software Leica Application Suite (LAS, 4.12 version) X. Values were calculated as percentage of changes compared with the blank control (only medium), in mean ± SD, *n* = 4. * *p* < 0.05; compared with the control group.

**Figure 8 molecules-30-02345-f008:**
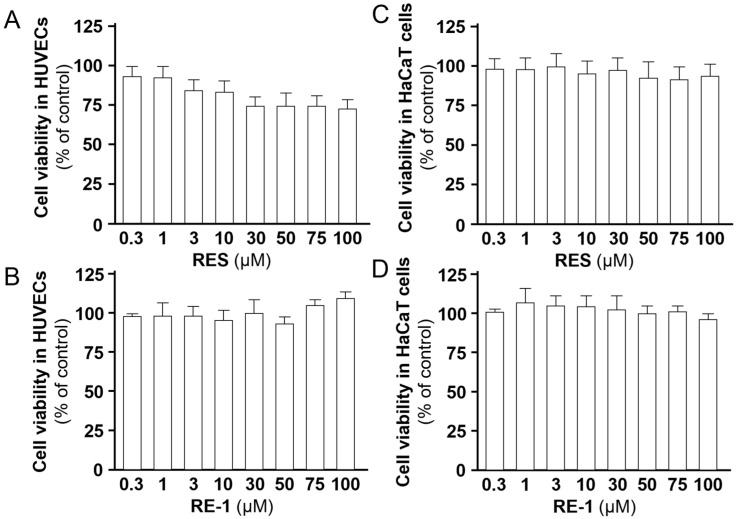
(**A**,**B**) Cell viabilities of RES and RE-1 in HUVECs. (**C**,**D**) Cell viabilities of RES and RE-1 in HaCaT cells. The cells were seeded onto 96-well plates for 24 h. After drug treatments, an MTT solution at a final concentration of 0.5 mg/mL was added to each well. Values were calculated as the percentage of changes compared with the blank control (only medium in use), in mean ± SD, *n* = 4.

**Table 1 molecules-30-02345-t001:** The transepithelial permeabilities of analytes were measured by mass spectrometry. Values were calculated as percentage of changes compared with the blank control (only medium in use), in mean ± SD, *n* = 4. ^a^ *P*_app_ value was calculated through the following equation: $P_{\mathrm{app}}=\left(\frac{dQ}{dt}\right)\times \frac{1}{C}\times \frac{1}{A}$; ^b^ standard deviation (SD) was calculated from four replicates; ^c^ sample tested at a concentration of 3 µM; ^d^ sample tested at a concentration of 0.3 µM.

Analytes	*P*_app_ Values (×10^−6^ cm/s) ^a^	SD ^b^
RES ^c^	18.2	1.3
RES ^d^	15.1	0.8
RE-1 ^c^	20.9	1.6
RE-1 ^d^	15.8	1.2

**Table 2 molecules-30-02345-t002:** The stability test was validated by determining of the intra- and inter-day variability through HPLC analysis. ^a,b^ The intra- and inter-day stability tests were determined by analyzing six replicates of the standard solution of the two analytes during a single day and six replicates of the samples examined on five successive days, respectively; ^c^ mean peak area was calculated from the average number of six replicates.

Analytes	Precision			
	Intra-day (*n* = 6) ^a^		Inter-day (*n* = 6) ^b^	
	Mean Peak Area ^c^	RSD (%)	Mean Peak Area ^c^	RSD (%)
RE-1 (acidic)	1316.9	5.84	1403.9	8.83
RE-1 (neutral)	1002.8	4.93	1046.6	9.21
RE-1 (basic)	755.8	5.81	747.2	9.35

## Data Availability

All data are available from the corresponding author upon reasonable request.

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
