# Peer review of "Design, Synthesis, and Biological Evaluations of a Novel Resveratrol-Type Analogue Against VEGF"

_molecules, 2025, doi:10.3390/molecules30112345_

Round 1

Reviewer 1 Report

Comments and Suggestions for Authors

The manuscript Molecules-3659954 is of merit and interest since it provides a novel and well-conducted series of experiments aimed to demonstrate the ability of Resveratrol and its synthetic analog RE-1 to bind VEGF preventing the activation of its receptor VEGFR and therefore the signaling cascade triggered by them, implicated in important biological functions such as cell proliferation, migration growth and apoptosis. The overactivation of this pathway is related to diseases such as cancer and wet age-related macular degeneration (wAMD), therefore the molecules able to block this pathway such as RES and RE-1 could be a potential therapeutic agent to treat or prevent the forehead mentioned diseases. Besides its therapeutic potency the main advantage of RES and RE-1 relies on its harmless activity to the organism. The authors employ a plethora of experimental techniques to carry out its research which provides a robust and well-presented set of results supporting the ability of RES and mainly RE-1 to prevent VEGF-mediated diseases. I would include in the manuscript the figures of cell viability instead of keeping them in the supplementary material and I highly recommend the paper of Lin et al for its publication in Molecules with minor changes.

The minor changes to improve the manuscript are reported below:

Line 18: in vitro and in vivo ITALIC

Line 19: remove and to make sense (The blockage of VEGF functions is due to its direct binding, and which subsequently enabled the analysis of structural-activity-relationship (SAR) around this hit molecule

Line 22: remove two hyphens

Line 40: remove comma after e.g.

Line 228: change etc by among others

Line 311: as well as you indicate the type of cells of HaCaT, do the same with HUVECs and Caco-2

Line 357: how long were the cells incubated with DCFH-DA? Did you do any washing step? Please explain this protocol with more detail.

Line 359: microscope settings such as magnification

In 4.5 and 4.9 sections you should explain the conditions of high and low control

Author Response

  1. “I would include in the manuscript the figures of cell viability instead of keeping them in the supplementary material…”

Our response: Thank you for the comment. We have removed the figure regarding cell viability from supplementary material to the main text of manuscript, i.e. Figure 8.

  1. “Line 18: in vitro and in vivo ITALIC; Line 19: remove and to make sense (The blockage of VEGF functions is due to its direct binding, and which subsequently enabled the analysis of structural-activity-relationship (SAR) around this hit molecule; Line 22: remove two hyphens; Line 40: remove comma after e.g.; Line 228: change etc by among others; Line 311: as well as you indicate the type of cells of HaCaT, do the same with HUVECs and Caco-2”.

Our response: Thank you for pointing out the mistake. We have revised the manuscript as suggested by the reviewer. Please see Line 18, 19, 20, 230 and 312-313.

  1. “Line 357: how long were the cells incubated with DCFH-DA? Did you do any washing step? Please explain this protocol with more detail; Line 359: microscope settings such as magnification.”

Our response: The cells were incubated with DCFH-DA for 20 min at the 37 °C incubator. Following the incubation, the cells were washed with PBS. Before being mounted with DAPI mountant, the cells were fixed with 4% paraformaldehyde in PBS for 15 min at room temperature. Other than that, the DCFH-DA-probed cells were photographed using a Leica SP8 confocal system at 20X magnification. We have included the information in the experimental section. Please see Line 359-368.

  1. In 4.5 and 4.9 sections you should explain the conditions of high and low control.

Our response: Thank you for the comment. We have revised the experimental protocols in both sections 4.5 and 4.9. Please see Line 344-345, 391-392.

Reviewer 2 Report

Comments and Suggestions for Authors

In this manuscript, the authors described the characterization of resveratrol derrivatives, RE-1 and RES in vitro. Both compounds showed target engagement in computational studies and SPR. The efficacy of both leads were assessed and validated in extensive cell assays. This manuscript is recommended for publication with the following minor revision:

  1. It is recommended that the authors show raw SPR binding curve along with the processed one. It is also nice to have negative and positive controls for this experiment.
  2. In the experimental section, the authors did not mention DMSO percentage in SPR studies. Can authors elaborate on this?
  3. It is recommended that the authors include loading control (GAPDH or β-actin) for all WB studies.
  4. The anti-proliferation effects in HUVEC cells could also be due to cytotoxicity. Can authors include toxicity data in other cells as well as during IF studies.

Author Response

  1. “It is recommended that the authors show raw SPR binding curve along with the processed one. It is also nice to have negative and positive controls for this experiment

Our response: Thank you for the comment. We have added the SPR raw data in the supplementary materials, i.e. Figure S2. A blank group (without drug treatment) was employed as a blank control.  

  1. In the experimental section, the authors did not mention DMSO percentage in SPR studies. Can authors elaborate on this.

Our response: We have included the DMSO percentage in the experimental section. Please see line 323.

  1. It is recommended that the authors include loading control (GAPDH or β-actin) for all WB studies.

Our response: Thank you for the recommendation. In this experiment, we have detected all phospho-signals and normalized to total protein (both ERK and VEGFR2), and the results were consistent across replicates. Due to time constraint, we are unbale to specifically include loading control (GAPDH or β-actin) this time, and we shall surely consider including additional loading controls in future studies.

  1. The anti-proliferation effects in HUVEC cells could also be due to cytotoxicity. Can authors include toxicity data in other cells as well as during IF studies

Our response: Thank you for the comment. We are aware of the fact that the anti-proliferation effects could result from cytotoxicity, and therefore we conducted MTT assays in both HUVECs and HaCaT cells to determine the cytotoxicity effects of resveratrol and RE-1. As shown in Figure 8, no significant apoptosis was observed from both resveratrol and RE-1 at relatively low concentrations, indicating that both chemicals contained very little cytotoxicity.